# Reject option models comprising out-of-distribution detection

## Abstract

The optimal prediction strategy for out-of-distribution (OOD) setups is a fundamental question in machine learning. In this paper, we address this question and present several contributions. We propose three reject option models for OOD setups: the Cost-based model, the Bounded TPR-FPR model, and the Bounded Precision-Recall model. These models extend the standard reject option models used in non-OOD setups and define the notion of an optimal OOD selective classifier. We establish that all the proposed models, despite their different formulations, share a common class of optimal strategies. Motivated by the optimal strategy, we introduce double-score OOD methods that leverage uncertainty scores from two chosen OOD detectors: one focused on OOD/ID discrimination and the other on misclassification detection. The experimental results consistently demonstrate the superior performance of this simple strategy compared to state-of-the-art methods. Additionally, we propose novel evaluation metrics derived from the definition of the optimal strategy under the proposed OOD rejection models. These new metrics provide a comprehensive and reliable assessment of OOD methods without the deficiencies observed in existing evaluation approaches.

## 1 Introduction

Most methods for learning predictors from data are based on the closed-world assumption, i.e., the training and the test samples are generated i.i.d. from the same distribution, so-called in-distribution (ID). However, in real-world applications, ID test samples can be contaminated by samples from another distribution, the so-called Out-of-Distribution (OOD), which is not represented in training examples. A trustworthy prediction model should detect OOD samples and reject to predict them, while simultaneously minimizing the prediction error on accepted ID samples.

In recent years, the development of deep learning models for handling OOD data has emerged as a critical challenge in the field of machine learning, leading to an explosion of research papers dedicated to developing effective OOD detection methods (OODD) [10, 11, 4, 3, 12, 8, 1, 17, 16, 19, 20]. Existing methods use various principles to learn a classifier of ID samples and a selective function that accepts the input for prediction or rejects it to predict. We further denote the pair of ID classifier and the selective function as OOD selective classifier, borrowing terminology from the non-OOD setup [7]. There is an agreement that a good OOD selective classifier should reject OOD samples and simultaneously achieve high classification accuracy on ID samples that are accepted [22]. To our knowledge, there is surprisingly no formal definition of an optimal OOD selective classifier. Consequently, there is also no consensus on how to evaluate the OODD methods. The commonly used metrics [21] evaluate only one aspect of the OOD selective classifier, either the accuracy of the ID classifier or the performance of the selective function as an OOD/ID discriminator. Such evaluation is inconclusive and usually inconsistent; e.g., the two most commonly used metrics, AUROC and OSCR, often lead to a completely reversed ranking of evaluated methods (see Sec. 3.4).

Submitted to 37th Conference on Neural Information Processing Systems (NeurIPS 2023). Do not distribute.

In this paper, we ask the following question: What would be the optimal prediction strategy for the OOD setup in the ideal case when ID and OOD distributions were known? To this end, we offer the contributions: (i) We propose three reject option models for the OOD setup: Cost-based model, bounded TPR-FPR model, and Bounded Precision-Recall model. These models extend the standard rejection models used in the non-OOD setup [2, 15] and define the notion of an optimal OOD classifier. (ii) We establish that all the proposed models, despite their different formulations, share a common class of optimal strategies. The optimal OOD selective classifier combines a Bayes ID classifier with a selective function based on a linear combination of the conditional risk and likelihood ratio of the OOD and ID samples. This selective function enables a trade-off between distinguishing ID from OOD samples and detecting misclassifications. (iii) Motivated by the optimal strategy, we introduce double-score OOD methods that leverage uncertainty scores from two chosen OOD detectors: one focused on OOD/ID discrimination and the other on misclassification detection. We show experimentally that this simple strategy consistently outperforms the state-of-the-art. (iv) We review existing metrics for evaluation of OODD methods and show that they provide incomplete view, if used separately, or inconsistent view of the evaluated methods, if used together. We propose novel evaluation metrics derived from the definition of optimal strategy under the proposed OOD rejection models. These new metrics provide a comprehensive and reliable assessment of OODD methods without the deficiencies observed in existing approaches.

## 2 Reject option models for OOD setup

The terminology of ID and OOD samples comes from the setups when the training set contains only ID samples, while the test set contains a mixture of ID and OOD samples. In this paper, we analyze which prediction strategies are optimal on the test samples, but we do not address the problem of learning such strategy. We follow the OOD setup from [5]. Let $\mathcal{X}$ be a set of observable inputs (or features), and $\mathcal{Y}$ a finite set of labels that can be assigned to in-distribution (ID) inputs. ID samples $(x, y) \in \mathcal{X} \times \mathcal{Y}$ are generated from a joint distribution $p_I \colon \mathcal{X} \times \mathcal{Y} \to \mathbb{R}_+$. Out-of-distribution (OOD) samples $x$ are generated from a distribution $p_O \colon \mathcal{X} \to \mathbb{R}_+$. ID and OOD samples share the same input space $\mathcal{X}$. Let $\emptyset$ be a special label to mark the OOD sample. Let $\bar{\mathcal{Y}} = \mathcal{Y} \cup \{\emptyset\}$ be an extended set of labels. In the testing stage the samples $(x, \bar{y}) \in \mathcal{X} \times \bar{\mathcal{Y}}$ are generated from the joint distribution $p \colon \mathcal{X} \times \bar{\mathcal{Y}} \to \mathbb{R}_+$ defined as a mixture of ID and OOD:

$$p(x, \bar{y}) = \left\{ \begin{array}{rl} p_O(x)\,\pi & \text{if } \bar{y} = \emptyset \\ p_I(x, \bar{y})\,(1 - \pi) & \text{if } \bar{y} \in \mathcal{Y} \end{array} \right., \tag{1}$$

where $\pi \in [0, 1)$ is the probability of observing the OOD sample. Our OOD setup subsumes the standard non-OOD setup as a special case when $\pi = 0$, and the reject option models that will be introduced below will become for $\pi = 0$ the known reject option models for the non-OOD setup.

Our goal is to design OOD selective classifier $q \colon \mathcal{X} \to \mathcal{D}$, where $\mathcal{D} = \mathcal{Y} \cup \{\text{reject}\}$, which either predicts a label, $q(x) \in \mathcal{Y}$, or it rejects the prediction, $q(x) = \text{reject}$, when (i) input $x \in \mathcal{X}$ prevents accurate prediction of $y \in \mathcal{Y}$ because it is noisy, or (ii) comes from OOD. We represent the selective classifier by the ID classifier $h \colon \mathcal{X} \to \mathcal{Y}$, and a stochastic selective function $c \colon \mathcal{X} \to [0, 1]$ that outputs a probability that the input is accepted [7], i.e.,

$$q(x) = (h, c)(x) = \left\{ \begin{array}{ll} h(x) & \text{with probability } c(x) \\ \text{reject} & \text{with probability } 1 - c(x) \end{array} \right.. \tag{2}$$

In the following sections, we propose three reject option models that define the notion of the optimal OOD selective classifier of the form (2) applied to samples generated by (1).

### 2.1 Cost-based rejection model for OOD setup

A classical approach to define an optimal classifier is to formulate it as a loss minimization problem. This requires defining a loss $\bar{\ell} \colon \bar{\mathcal{Y}} \times \mathcal{D} \to \mathbb{R}_+$ for each combination of the label $\bar{y} \in \bar{\mathcal{Y}} = \mathcal{Y} \cup \{\emptyset\}$ and the output of the classifier $q(x) \in \mathcal{D} = \mathcal{Y} \cup \{\text{reject}\}$. Let $\ell \colon \mathcal{Y} \times \mathcal{Y} \to \mathbb{R}_+$ be some application-specific loss on ID samples, e.g., 0/1-loss or MAE. Furthermore, we need to define the loss for the case where the input is OOD sample $\bar{y} = \emptyset$ or the classifier rejects $q(x) = \text{reject}$. Let $\varepsilon_1 \in \mathbb{R}_+$ be the loss for rejecting the ID sample, $\varepsilon_2 \in \mathbb{R}_+$ loss for prediction on the OOD sample, and $\varepsilon_3 \in \mathbb{R}_+$ loss for correctly rejecting the OOD sample. $\ell, \varepsilon_1, \varepsilon_2$ and $\varepsilon_3$ can be arbitrary, but we assume that

$\varepsilon_2 > \varepsilon_3$. The loss $\bar{\ell}$ is then:

$$\bar{\ell}(\bar{y}, q) = \left\{ \begin{array}{rll} \ell(\bar{y}, q) & \text{if} & \bar{y} \in \mathcal{Y} \wedge q \in \mathcal{Y} \\ \varepsilon_1 & \text{if} & \bar{y} \in \mathcal{Y} \wedge q = \text{reject} \\ \varepsilon_2 & \text{if} & \bar{y} = \emptyset \wedge q \in \mathcal{Y} \\ \varepsilon_3 & \text{if} & \bar{y} = \emptyset \wedge q = \text{reject} \end{array} \right. \tag{3}$$

Having the loss $\bar{\ell}$, we can define the optimal OOD selective classifier as a minimizer of the expected risk $R(h, c) = \mathbb{E}_{x, y \sim p(x, \bar{y})} \bar{\ell}(\bar{y}, (h, c)(x))$.

**Definition 1** *(Cost-based OOD model) An optimal OOD selective classifier $(h_C, c_C)$ is a solution to the minimization problem $\min_{h,c} R(h, c)$ where we assume that both minimizers exist.*

An optimal solution of the cost-based OOD model requires three components: The Bayes ID classifier

$$h_B(x) \in \operatorname*{Argmin}_{y' \in \mathcal{Y}} \sum_{y \in \mathcal{Y}} p_I(y \mid x) \ell(y, y'), \tag{4}$$

its conditional risk $r_B(x) = \sum_{y \in \mathcal{Y}} p_I(y \mid x) \ell(y, h_B(x))$, and the likelihood ratio of the OOD and ID inputs, $g(x) = \frac{p_O(x)}{p_I(x)}$, which we defined to be $g(x) = \infty$ for $p_I(x) = 0$.

**Theorem 1** *An optimal selective classifier $(h_C, c_C)$ under the cost-based OOD model is composed of the Bayes classifier (4), $h_C = h_B$, and the selective function*

$$c_C(x) = \left\{ \begin{array}{lll} 1 & \textit{if} & s_C(x) < \varepsilon_1 \\ \tau & \textit{if} & s_C(x) = \varepsilon_1 \\ 0 & \textit{if} & s_C(x) > \varepsilon_1 \end{array} \right. \quad \textit{using the score} \quad s_C(x) = r_B(x) + (\varepsilon_2 - \varepsilon_3) \frac{\pi}{1 - \pi} g(x) \tag{5}$$

*where $\tau$ is an arbitrary number in $[0, 1]$, and $\varepsilon_1$, $\varepsilon_2$, $\varepsilon_3$ are losses defining the extended loss (3).*

Note that $\tau$ can be arbitrary and therefore a deterministic selective function $c_C(x) = [\![s_C(x) \leq \varepsilon_1]\!]$ is also optimal. An optimal selective function accepts inputs based on the score $s_C(x)$, which is a linear combination of two functions, conditional risk $r_B(x)$ and the likelihood ratio $g(x) = p_O(x)/p_I(x)$.

**Relation to cost-based model for Non-OOD setup** For $\pi = 0$, the cost-based OOD model reduces to the standard cost-based model of the reject option classifier in a non-OOD setup [2]. In the non-OOD setup, we do not need to specify the losses $\varepsilon_2$ and $\varepsilon_3$ and the risk $R(h, c)$ simplifies to $R'(h, c) = \mathbb{E}_{x, y \sim p_I(x, y)} \big[ \ell(y, h(x)) c(x) + \varepsilon_1 (1 - c(x)) \big]$. The well-known optimal solution is composed of the Bayes classifier $h_B(x)$ as in the OOD case; however, the selection function $c'_C(x) = [\![r(x) \leq A]\!]$ accepts the input solely based on the conditional risk $r(x)$.

## 2.2 Bounded TPR-FPR rejection model

The cost-based OOD model requires the classification loss $\ell$ for ID samples and defining the costs $\varepsilon_1$, $\varepsilon_2$, $\varepsilon_3$ which is difficult in practice because the physical units of $\ell$ and $\varepsilon_1, \varepsilon_2, \varepsilon_3$ are often different. In this section, we propose an alternative approach which requires only the classification loss $\ell$ while costs $\varepsilon_1, \varepsilon_2, \varepsilon_3$ are replaced by constraints on the performance of the selective function.

The selective function $c \colon \mathcal{X} \to [0, 1]$ can be seen as a discriminator of OOD/ID samples. Let us consider ID and OOD samples as positive and negative classes, respectively. We introduce three metrics to measure the performance of the OOD selective classifier $(h, c)$. We measure the performance of selective function by the True Positive Rate (TPR) and the False Positive Rate (FPR). The TPR is defined as the probability that ID sample is accepted by the selective function $c$, i.e.,

$$\phi(c) = \int_{\mathcal{X}} p(x \mid \bar{y} \neq \emptyset) c(x) \, dx = \int_{\mathcal{X}} p_I(x) c(x) \, dx. \tag{6}$$

The FPR is defined as the probability that OOD sample is accepted by the selective function $c$, i.e.,

$$\rho(c) = \int_{\mathcal{X}} p(x \mid \bar{y} = \emptyset) c(x) \, dx = \int_{\mathcal{X}} p_O(x) c(x) \, dx. \tag{7}$$

The second identity in (6) and (7) is obtained after substituting the definition of $p(x, \bar{y})$ from (1). Lastly, we characterize the performance of the ID classifier $h\colon \mathcal{X} \to \mathcal{Y}$ by the selective risk

$$\mathrm{R}^{\mathrm{S}}(h, c) = \frac{\int_{\mathcal{X}} \sum_{y \in \mathcal{Y}} p_I(x, y)\, \ell(h(x), y)\, c(x)\ dx}{\phi(c)}$$

defined for non-zero $\phi(c)$, i.e., the expected loss of the classifier $h$ calculated on the ID samples accepted by the selective function $c$.

**Definition 2 (Bounded TPR-FPR model)** *Let $\phi_{\min} \in [0, 1]$ be the minimal acceptable TPR and $\rho_{\max} \in [0, 1]$ maximal acceptable FPR. An optimal OOD selective classifier $(h_T, c_T)$ under the bounded TPR-FPR model is a solution of the problem*

$$\min_{h \in \mathcal{Y}^{\mathcal{X}}, c \in [0,1]^{\mathcal{X}}} \mathrm{R}^{\mathrm{S}}(h, c) \qquad s.t. \qquad \phi(c) \geq \phi_{\min} \quad and \quad \rho(c) \leq \rho_{\max}\,, \tag{8}$$

*where we assume that both minimizers exist.*

**Theorem 2** *Let $(h, c)$ be an optimal solution to (8). Then $(h_B, c)$, where $h_B$ is the Bayes ID classifier (4), is also optimal to (8).*

According to Theorem 2, the Bayes ID classifier $h_B$ is an optimal solution to (8) that defines the bounded TPR-FPR model. This is not surprising, but it is a practically useful result, because it allows one to solve (8) in two consecutive steps: First, set $h_T$ to the Bayes ID classifier $h_B$. Second, when $h_T$ is fixed, the optimal selection function $c_T$ is obtained by solving (8) only w.r.t. $c$ which boils down to:

**Problem 1 (Bounded TPR-FPR model for known $h(x)$)** *Given ID classifier $h\colon \mathcal{X} \to \mathcal{Y}$, the optimal selective function $c^*\colon \mathcal{X} \to [0, 1]$ is a solution to*

$$\min_{c \in [0,1]^{\mathcal{X}}} \mathrm{R}^{\mathrm{S}}(h, c) \qquad s.t. \qquad \phi(c) \geq \phi_{\min}\,, \quad and \quad \rho(c) \leq \rho_{\max}\,.$$

Problem 1 is meaningful even if $h$ is not the Bayes ID classifier $h_B$. We can search for an optimal selective function $c^*(x)$ for any fixed $h$, which in practice is usually our best approximation of $h_B$ learned from the data.

**Theorem 3** *Let $h\colon \mathcal{X} \to \mathcal{Y}$ be ID classifier and $r\colon \mathcal{X} \to \mathbb{R}$ its conditional risk $r(x) = \sum_{y \in \mathcal{Y}} p_I(y \mid x)\ell(y, h(x))$. Let $g(x) = p_I(x)/p_I(x)$ be the likelihood ratio of ID and OOD samples. Then, the set of optimal solutions of Problem 1 contains the selective classifier*

$$c^*(x) = \begin{cases} 0 & if \quad s(x) > \lambda \\ \tau(x) & if \quad s(x) = \lambda \\ 1 & if \quad s(x) < \lambda \end{cases} \qquad using\ score \quad s(x) = r(x) + \mu\, g(x) \tag{9}$$

*where decision threshold $\lambda \in \mathbb{R}$, and multiplier $\mu \in \mathbb{R}$ are constants and $\tau\colon \mathcal{X} \to [0, 1]$ is a function implicitly defined by the problem parameters.*

The optimal $c^*(x)$ is based on the score composed of a linear combination of $r(x)$ and $g(x)$ as in the case of the cost-based model (5). Unlike the cost-based model, the acceptance probability $\tau(x)$ for boundary inputs $\mathcal{X}_{s(x)=\lambda} = \{x \in \mathcal{X} \mid s(x) = \lambda\}$ cannot be arbitrary, in general. However, if $\mathcal{X}$ is continuous, the set $\mathcal{X}_{s(x)=\lambda}$ has probability measure zero, up to some pathological cases, and $\tau(x)$ can be arbitrary, i.e., the deterministic $c^*(x) = [\![ s(x) \leq \lambda ]\!]$ is optimal. If $\mathcal{X}$ is finite, the value of $\tau(x)$ can be found by linear programming. The linear program and more details on the form of $\tau(x)$ are in the Appendix.

**Relation to Bounded-Abstention model for the non-OOD setup**   For $\pi = 0$, the bounded TPR-FPR model reduces to the bounded-abstention option model for non-OOD setup [15]. Namely, $\rho(c) \leq \rho_{\max}$ can be removed because there are no OOD samples, and (8) becomes the bounded-abstention model: $\min_{h,c} \mathrm{R}^{\mathrm{S}}(h, c)$, s.t. $\phi(c) \geq \phi_{\min}$, which seeks the selective classifier with guaranteed TPR and minimal selective risk. In the non-OOD setup, TPR is called *coverage*. An optimal solution of the bounded abstention model [6], is composed of the Bayes ID classifier $h_B$, and the same optimal selective function as the TPR-FPR model (9), however, with $\mu = 0$ and $\tau(x) = \tau$, $\forall x \in \mathcal{X}$, i.e., the score depends only on $r(x)$ and an identical randomization is applied in all edge cases [6]. Therefore, $r(x)$ is the optimal score to detect misclassified ID samples in non-OOD setup as it allows to achieve the minimal selective risk $\mathrm{R}^{\mathrm{S}}$ for any fixed coverage (TPR, $\phi$).

## 2.3 Bounded Precision-Recall rejection model

The optimal selective classifier under the bounded TPR-FPR model does not depend on the prior of the OOD samples $\pi$, which is useful, e.g., when $\pi$ is unknown in the testing stage. In the case $\pi$ is known, it might be more suitable to constrain the precision rather than the FPR, while the constraint on TPR remains the same. In the context of precision, we denote $\phi(c)$ as recall instead of TPR. The precision $\kappa(c)$ is defined as the portion of samples accepted by $c(x)$ that are actual ID samples, i.e.,

$$\kappa(c) = \frac{(1-\pi)\int_{\mathcal{X}} p(x \mid \bar{y} \neq \emptyset)\, c(x)\, dx}{\int_{\mathcal{X}} p(x)\, c(x)\, dx} = \frac{(1-\pi)\,\phi(c)}{\rho(c)\,\pi + \phi(c)\,(1-\pi)}\,.$$

**Definition 3 (Bounded Precision-Recall model)** *Let $\kappa_{\min} \in [0,1]$ be a minimal acceptable precision and $\phi_{\min} \in [0,1]$ minimal acceptable recall (a.k.a. TPR). An optimal selective classifier $(h_P, c_P)$ under the bounded Precision-Recall model is a solution of the problem*

$$\min_{h \in \mathcal{Y}^{\mathcal{X}}, c \in [0,1]^{\mathcal{X}}} \mathrm{R}^{\mathrm{S}}(h,c) \quad s.t. \quad \phi(c) \geq \phi_{\min} \quad and \quad \kappa(c) \geq \kappa_{\min} \tag{10}$$

*where we assume that both minimizers exist.*

**Theorem 4** *Let $(h,c)$ be an optimal solution to (10). Then $(h_B, c)$, where $h_B$ is the Bayes ID classifier (4), is also optimal to (10).*

Theorem 4 ensures that the Bayes ID classifier is an optimal solution to (10). After fixing $h_P = h_B$, the search for an optimal selective function $c$ leads to:

**Problem 2 (Bounded Prec-Recall model for known $h(x)$)** *Given ID classifier $h\colon \mathcal{X} \to \mathcal{Y}$, the optimal selective function $c^*\colon \mathcal{X} \to [0,1]$ is a solution to*

$$\min_{c \in [0,1]^{\mathcal{X}}} \mathrm{R}^{\mathrm{S}}(h,c) \quad s.t. \quad \phi(c) \geq \phi_{\min} \quad and \quad \kappa(c) \geq \kappa_{\min}\,.$$

**Theorem 5** *Let $h\colon \mathcal{X} \to \mathcal{Y}$ be ID classifier and $r\colon \mathcal{X} \to \mathbb{R}$ its conditional risk $r(x) = \sum_{y \in \mathcal{Y}} p_I(y \mid x)\ell(y, h(x))$. Let $g(x) = p_O(x)/p_I(x)$ be the likelihood ratio of OOD and ID samples. Then, the set of optimal solutions of Problem 2 contains the selective function*

$$c^*(x) = \begin{cases} 0 & if \quad s(x) > \lambda \\ \tau(x) & if \quad s(x) = \lambda \\ 1 & if \quad s(x) < \lambda \end{cases} \quad using\ the\ score \quad s(x) = r(x) + \mu\, g(x) \tag{11}$$

*where detection threshold $\lambda \in \mathbb{R}$, and multiplier $\mu \in \mathbb{R}$ are constants and $\tau\colon \mathcal{X} \to [0,1]$ is a function implicitly defined by the problem parameters.*

## 2.4 Summary

We proposed three rejection models for OOD setup which define the notion of optimal OOD selective classifier: Cost-based model, Bounded TRP-FPR model, and Bounded Precision-Recall model. We established that all three models, despite different formulation, share the class of optimal prediction strategies. Namely, the optimal OOD selective classifier $(h^*, c^*)$ is composed of the Bayes ID classifier (4), $h^* = h_B$, and the selective function

$$c^*(x) = \begin{cases} 0 & if \quad s(x) > \lambda \\ \tau(x) & if \quad s(x) = \lambda \\ 1 & if \quad s(x) < \lambda \end{cases} \quad where \quad s(x) = r(x) + \mu\, g(x) \tag{12}$$

where $\lambda$, $\mu$, and $\tau(x)$ are specific for the used rejection model. However, in all cases, the optimal uncertainty score $s(x)$ for accepting the inputs is based on a linear combination of the conditional risk $r(x)$ of the ID classifier $h^*$ and the OOD/ID likelihood ratio $g(x) = p_O(x)/p_I(x)$. On the other hand, from the optimal solution of the well-known Neyman-Person problem [14], it follows that the likelihood ratio $g(x)$ is the optimal score of OOD/ID discrimination. Our results thus show that the optimal OOD selective function needs to trade-off the ability to detect the misclassification of ID samples and the ability to distinguish ID from OOD samples.

**Single-score vs. double-score OODD methods** The existing OODD methods, which we further call *single-score methods*, produce a classifier $h\colon \mathcal{X} \to \mathcal{Y}$ and an uncertainty score $s\colon \mathcal{X} \to \mathbb{R}$. The score $s(x)$ is used to construct a selective function $c(x) = [\![s(x) \leq \lambda]\!]$ where $\lambda \in \mathbb{R}$ is a decision threshold chosen in post-hoc evaluation. Hence, the existing methods effectively produce a set of selective classifiers $\mathcal{Q} = \{(h, c) \mid c(x) = [\![s(x) \leq \lambda]\!], \lambda \in \mathbb{R}\}$. In contrast to existing methods, we established that the optimal selective function is always based on a linear combination of two scores: conditional risk $r(x)$ and likelihood ratio $g(x)$. Therefore, we propose the *double-score method*, which in addition to a classifier $h(x)$, produces two scores, $s_r\colon \mathcal{X} \to \mathbb{R}$ and $s_g\colon \mathcal{X} \to \mathbb{R}$, and uses their combination $s(x) = s_r(x) + \mu\, s_g(x)$ to accept inputs. Formally, the double-score method produces a set of selective classifiers $\mathcal{Q} = \{(h, c) \mid c(x) = [\![s_r(x) + \mu\, s_g(x) \leq \lambda]\!], \mu \in \mathbb{R}, \lambda \in \mathbb{R}\}$. The double-score strategy can be used to leverage uncertainty scores from two chosen OODD methods: one focused on OOD/ID discrimination and the other on misclassification detection.

# 3 Post-hoc tuning and evaluation metrics

Let $\mathcal{T} = ((x_i, \bar{y}_i) \in \mathcal{X} \times \bar{\mathcal{Y}} \mid i = 1, \ldots, n)$ be a set of validation examples i.i.d. drawn from a distribution $p(x, \bar{y})$. Given a set of selective classifiers $\mathcal{Q}$, trained by the single-score or double-score OODD method, the goal of the post-hoc tuning is to use $\mathcal{T}$ to select the best selective classifier $(h_n, c_n) \in \mathcal{Q}$ and estimate its performance on unseen samples generated from the same $p(x, \bar{y})$. This task requires a notion of an optimal selective classifier which we defined by the proposed rejection models. In Sec 3.2 and Sec 3.3, we propose the post-hoc tuning and evaluation metrics for the Bounded TPR-FPR and Bounded Precision-Recall models, respectively. In Sec 3.4 we review the existing evaluation metrics for OODD methods and point out their deficiencies. We will exemplify the proposed metrics on synthetic data and OODD methods described in Sec 3.1.

## 3.1 Synthetic data and exemplar single-score and double-score OODD methods

Let us consider a simple 1-D setup. The input space is $\mathcal{X} = \mathbb{R}$ and there are three ID labels $\mathcal{Y} = \{1, 2, 3\}$. ID samples are generated from $p_I(x, 1) = 0.3\mathcal{N}(x; -1, 1)$, $p_I(x, 2) = 0.3\mathcal{N}(x; 1, 1)$, $p_I(x, 3) = 0.4\mathcal{N}(x; 3, 1)$, where $\mathcal{N}(x; \mu, \sigma)$ is normal distribution with mean $\mu$ and variance $\sigma$. OOD is the normal distribution $p_O(x) = \mathcal{N}(x; 3, 0.2)$, and the OOD prior $\pi = 0.25$. We use $0/1$-loss $\ell(y, y') = [\![y \neq y']\!]$, i.e., $\mathrm{R}^{\mathrm{S}}$ is the classification error on accepted inputs. The known ID and OOD allows us to evaluate the Bayes ID classifier $h_B(x)$ by (4), its conditional risk $r_B(x) = \min_{y' \in \mathcal{Y}} \sum_{y \in \mathcal{Y}} p_I(y \mid x)\ell(y, y')$ and the OOD/ID likelihood ratio $g(x) = p_O(x)/p_I(x)$.

We consider 3 exemplar single-score OODD methods A, B, C. The methods produce the same optimal classifier $h^*(x)$ and the selective functions $c(x) = [\![r_B(x) + \mu\, g(x) \leq \lambda]\!]$ with a different setting of $\mu$. I.e., the method $k \in \{A, B, C\}$ produces the set of selective classifiers $\mathcal{Q}_k = \{(h^*(x), c(x)) \mid c(x) = [\![r_B(x) + \mu_k\, g(x) \leq \lambda]\!], \lambda \in \mathbb{R}\}$, where the constant $\mu_k$ is defined as follows:

- Method A($\infty$): $\mu = \infty$, $s(x) = g(x)$. This corresponds to the optimal OOD/ID discriminator.
- Method B(0.2): $\mu = 0.2$, $s(x) = r_B(x) + 0.2g(x)$. Combination of method A and C.
- Method C(0): $\mu = 0$, $s(x) = r_B(x)$. This corresponds to the optimal misclassification detector.

We also consider a double-score method, Method D($\mathbb{R}$), which outputs the same optimal classifier $h_*(x)$, and scores $s_r(x) = r(x)$ and $s_g(x) = g(x)$. I.e., Method D($\mathbb{R}$) produces the set of selective classifiers $\mathcal{Q}_D = \{(h^*(x), c(x)) \mid c(x) = [\![r(x) + \mu\, g(x) \leq \lambda]\!], \mu \in \mathbb{R}, \lambda \in \mathbb{R}\}$. Note that we have shown that $\mathcal{Q}_D$ contains an optimal selective classifier regardless of the reject option model used.

## 3.2 Bounded TPR-FPR rejection model

The bounded TPR-FPR model is defined using the selective risk $\mathrm{R}^{\mathrm{S}}(h, c)$, TPR $\phi(c)$ and FPR $\rho(c)$ the value of which can be estimated from the validation set $\mathcal{T}$ as follows:

$$\mathrm{R}_{\mathrm{n}}^{\mathrm{S}}(h, c) = \frac{\sum_{i \in \mathcal{I}_I} \ell(y_i, h(x_i))\, c(x_i)}{\sum_{i \in \mathcal{I}_I} c(x_i)}, \quad \phi_n(h, c) = \frac{1}{|\mathcal{I}_I|} \sum_{i \in \mathcal{I}_I} c(x_i), \quad \rho_n(h, c) = \frac{1}{|\mathcal{I}_O|} \sum_{i \in \mathcal{I}_O} c(x_i)$$

where $\mathcal{I}_I = \{i \in \{1, \ldots, n\} \mid \bar{y}_i \neq \emptyset\}$ and $\mathcal{I}_O = \{i \in \{1, \ldots, n\} \mid \bar{y}_i = \emptyset\}$ are indices of ID and OOD samples in $\mathcal{T}$, respectively.

| Method | Proposed metrics | | ↑ Existing metrics | | |
| | TPR-FPR model | Prec-Recall model | | | |
| | ↓ Selective risk at TPR(0.7),FPR(0.2) | ↓ Selective risk at Prec(0.9),Recall(0.7) | AUROC | AUPR | OSCR |
|---|---|---|---|---|---|
| A($\infty$) | 0.157 | 0.157 | **0.88** | **0.96** | 0.82 |
| B(0.2) | 0.143 | 0.143 | 0.86 | 0.95 | 0.83 |
| C(0) | unable | unable | 0.76 | 0.92 | **0.86** |
| D($\mathbb{R}$) proposed | **0.133** | **0.129** | **0.88** | **0.96** | **0.86** |

Table 1: Evalution of the examplar single-score methods A, B, C and the proposed double-score method D on synthetic data using the proposed metrics and the existing ones. The selective risk correponds to the classification error on accepted ID samples.

Given the target TPR $\phi_{min} \in (0, 1]$ and FPR $\rho_{max} \in (0, 1]$, the best selective classifier $(h_n, c_n)$ out of $\mathcal{Q}$ is found by solving:

$$(h_n, c_n) \in \underset{(h,c)\in\mathcal{Q}}{\mathrm{Argmin}}\, \mathrm{R}_n^S(h, c) \quad \text{s.t.} \quad \phi_n(h, c) \geq \phi_{min}, \quad \text{and} \quad \rho_n(h, c) \leq \rho_{max}. \quad (13)$$

**Proposed evaluation metric** If problem (13) is feasible, $\mathrm{R}_n^S(h_n, c_n)$ is reported as the performance estimator of OODD method producing $\mathcal{Q}$. Otherwise, the method is marked as unable to achieve the target TPR and FPR. Tab. 1 shows the selective risk for the methods A-D at the target TPR $\phi_{min} = 0.7$ and FPR $\rho_{max} = 0.2$. The minimal $\mathrm{R}_n^S$ is achieved by method D($\mathbb{R}$), followed by B(0.2) and A($\infty$), while C(0) is unable to achieve the target TPR and FPR. One can visualize $\mathrm{R}_n^S$ in a range of operating points while bounding only $\rho_{max}$ or $\phi_{min}$. E.g., by fixing $\rho_{max}$ we can plot $\mathrm{R}_n^S$ as a function of attainable values of $\phi_n$ by which we obtain the Risk-Coverage curve, known from non-OOD setup, at $\rho_{max}$. Recall that TPR is coverage. See Appendix for Risk-Coverage curve at $\rho_{max}$ for methods A-D.

**ROC curve** The problem (13) can be infeasible. To choose a feasible target on $\phi_{min}$ and $\rho_{max}$, it is advantageous to plot the ROC curve, i.e., values of TPR and FPR attainable by the classifiers in $\mathcal{Q}$. For single-score methods, the ROC curve is a set of points obtained by varying the decision threshold: $\mathrm{ROC}(\mathcal{Q}) = \{(\phi_n(h, c), \rho_n(h, c)) \mid c(x) = [\![s(x) \leq \lambda]\!], \lambda \in \mathbb{R}\}$. In case of double-score methods, we vary $\rho_{max} \in [0, 1]$ and for each $\rho_{max}$ we choose the maximal feasible $\phi_n$. I.e., ROC curve is $\mathrm{ROC}(\mathcal{Q}) = \{(\phi, \rho_{max}) \mid \phi = \max_{(h,c)\in\mathcal{Q}} \phi_n(h, c) \ s.t. \ \rho_n(h, c) \leq \rho_{max}, \quad \rho_{max} \in [0, 1]\}$. See Appendix for ROC curve of the methods A-D. In Tab. 1 we report the Area Under ROC curve (AUROC) which is a commonly used summary of the entire ROC curve. The highest AUROC achieved Methods A($\infty$) and E($\mathbb{R}$). Recall that Method A($\infty$) uses the optimal ID/OOD discriminator and the proposed Method E($\mathbb{R}$) subsumes A($\infty$).

### 3.3 Bounded Precision-Recall rejection model

Let $\kappa_n(c) = (1 - \pi)\,\phi_n(c)/((1 - \pi)\phi_n(c) + \pi\rho_n(c))$ be the sample precision of the selective function $c$. Given the target recall $\phi_{min} \in (0, 1]$ and precision $\kappa_{min} \in (0, 1]$, the best selective classifier $(h_n, c_n)$ out of $\mathcal{Q}$ is found by solving

$$(h_n, c_n) \in \underset{(h,c)\in\mathcal{Q}}{\mathrm{Argmin}}\, \mathrm{R}_n^S(h, c) \quad \text{s.t.} \quad \phi_n(h, c) \geq \phi_{min}, \quad \kappa_n(h, c) \geq \kappa_{min}. \quad (14)$$

**Proposed evaluation metric** If problem (14) is feasible, $\mathrm{R}_n^S(h_n, c_n)$ is reported as the performance estimator of OODD method which produced $\mathcal{Q}$. Otherwise, the method is marked as unable to achieve the target Precison/Recall. Tab. 1 shows the selective risk for the methods A-D at the Precision $\kappa_{min} = 0.9$ and recall $\phi_{max} = 0.7$. The minimal $\mathrm{R}_n^S$ is achieved by the proposed method D($\mathbb{R}$), followed by B(0.2) and A($\infty$), while method C(0) is unable to achieve the target Precision/Recall. Note that single-score methods A-C achieve the same $\mathrm{R}_n^S$ under both TPR-FPR and Prec-Recall models while the results for double-score method D($\mathbb{R}$) differ. The reason is that both models share the same constraint $\phi_n \geq 0.7$ (TPR is Recall) which is active, while the other two constraints are not active because $\mathrm{R}_n^S$ is a monotonic function w.r.t. the value of the decision threshold.

**Precision-Recall (PR) curve**    To choose feasible bounds on $\kappa_{min}$ and $\phi_{min}$ before solving (14), one can plot the PR curve, i.e., the values of precision and recall attainable by the classifiers in $\mathcal{Q}$. For single-score methods, the PR curve is a set of points obtained by varying the decision threshold: $\text{PR}(\mathcal{Q}) = \{(\kappa_n(h,c), \phi_n(h,c)) \mid c(x) = [\![s(x) \leq \lambda]\!], \lambda \in \mathbb{R}\}$. In case of double-score methods, we vary $\phi_{min} \in [0,1]$ and for each $\phi_{min}$ we choose the maximal feasible $\kappa_n$, i.e., $\text{PR}(\mathcal{Q}) = \{(\kappa, \phi_{min}) \mid \kappa = \max_{(h,c) \in \mathcal{Q}} \kappa_n(h,c) \ s.t. \ \phi_n(h,c) \geq \phi_{min}, \quad \phi_{min} \in [0,1]\}$. See Appendix for PR curve of the methods A-D. We compute the Area Under the PR curve and report it for Methods A-D in Tab. 1. Rankings of the methods w.r.t AUPR and AUROC are the same.

## 3.4  Shortcomings of existing evaluation metrics

The most commonly used metrics to evaluate OODD methods are the AUROC and AUPR [10, 13, 3, 12, 1, 16]. Both metrics measure the ability of the selective function $c(x)$ to distinguish ID from OOD samples. AUROC and AUPR are often the only metrics reported although they completely ignore the performance of the ID classifier. Our synthetic example shows that high AUROC/AUPR is not a precursor of a good OOD selective classifier. E.g., Method A($\infty$), using optimal OOD/ID discriminator, attains the highest (best) AUROC and AUPR (see Tab. 1), however, at the same time Method A($\infty$) achieves the highest (worst) $\text{R}_n^{\text{S}}$ under both rejection models, and it is also the worst misclassification detector according to the OSCR score defined below.

The performance of the ID classifier $h(x)$ is usually evaluated by the ID classification accuracy (a.k.a. closed set accuracy) [13, 3] and by the OSCR score [4, 8, 1]. The ID accuracy measures the performance of $h(x)$ assuming all inputs are accepted, i.e., $c(x) = 1, \forall x \in \mathcal{X}$, hence it says nothing about the performance on the actually accepted samples like $\text{R}_n^{\text{S}}$. E.g., Methods A-D in our synthetic example use the same classifier $h(x)$ and hence have the same ID accuracy, however, they perform quite differently in terms of the other more relevant metrics, like $\text{R}_n^{\text{S}}$ or OSCR. The OSCR score is defined as the area under CCR versus FPR curve [21], where the CCR stands for the correct classification rate on the accepted ID samples; in case of 0/1-loss $\text{CCR} = 1 - \text{R}_n^{\text{S}}$. The CCR-FPR curve evaluates the performance of the ID classifier on the accepted samples, but it ignores the ability of $c(x)$ to discriminate OOD and ID samples as it does not depend on TPR. E.g., Method D(0), using the optimal misclassification detector, achieves the highest (best) OSCR score; however, at the same time, it has the lowest (worst) AUROC and AUPR.

Other, less frequently used metrics involve: F1-score, FPR@TPRx, TNR@TPRx, CCR@FPRx [10, 8, 1, 21, 16]. All these metrics are derived from either ROC, PR or CCR-FPR curve, and hence they suffer with the same conceptual problems as AUROC, AUPR and OSCR, respectively.

We argue that the existing metrics evaluate only one aspect of the OOD selective classifier, namely, either the ability to discriminate ID from OOD samples, or the performance of ID classifier on the accepted (or on possibly all) ID samples. We show that in principle there can be methods that are best OOD/ID discriminators but the worst misclassification detectors and vice versa. Therefore, using individual metrics can (and often does) provide inconsistent ranking of the evaluated methods.

## 3.5  Summary

We propose novel evaluation metrics derived from the definition of the optimal strategy under the proposed OOD rejection models. The proposed metrics simultaneously evaluate the classification performance on the accepted ID samples and they guarantee the perfomance of the OOD/ID discriminator, either via constraints in TPR-FPR or Precision-Recall pair. Advantages of the proposed metrics come at a price. Namely, we need to specify feasible target TPR and FPR, or Precision and Recall, depending on the model used. However, feasible values of TPR-FPR and Prec-Recall pairs can be easily read out of the ROC and PR curve, respectively. We argue that setting these extra parameters is better than using the existing metrics that provide incomplete, if used separately, or inconsistent, if used in combination, view of the evaluated methods.

Another issue is solving the problems (13) and (14) to compute the proposed evaluation metrics and figures. Fortunately, both problems lead to optimization w.r.t one or two varibales in case of the single-score and double-score methods, respectively. A simple and efficient algorithm to solve the problems in $\mathcal{O}(n \log n)$ time is provided in Appendix.

| | OOD: notmnist | | | OOD: fashionmnist | | | OOD: cifar10 | | |
|---|---|---|---|---|---|---|---|---|---|
| Method | ↓ S. risk at TPR(0.80) FPR(0.08) | ↑ AUROC | ↑ OSCR | ↓ S. risk at TPR(0.80) FPR(0.10) | ↑ AUROC | ↑ OSCR | ↓ S. risk at TPR(0.80) FPR(0.29) | ↑ AUROC | ↑ OSCR |
| MSP [10] | **0.00014** | 0.936 | **0.996** | **0.00013** | 0.956 | **0.994** | **0.00013** | 0.989 | 0.991 |
| MLS [9] | 0.00139 | 0.941 | 0.993 | 0.00139 | 0.972 | 0.991 | 0.00139 | **0.993** | 0.990 |
| ODIN [11] | 0.00069 | 0.942 | 0.993 | 0.00069 | 0.970 | 0.991 | 0.00069 | 0.993 | 0.990 |
| REACT [17] | 0.00637 | 0.962 | 0.991 | 0.00637 | **0.985** | 0.990 | 0.00637 | 0.992 | 0.989 |
| KNN [19] | 0.00041 | 0.976 | 0.991 | 0.00041 | 0.947 | 0.993 | 0.00041 | 0.976 | 0.991 |
| VIM [20] | 0.00193 | **0.983** | 0.990 | 0.00194 | 0.926 | 0.993 | 0.00194 | 0.860 | **0.995** |
| KNN+MSP | **0.00000** | 0.976 | **0.996** | **0.00000** | 0.962 | **0.994** | **0.00000** | 0.991 | 0.991 |
| VIM+MSP | 0.00014 | **0.987** | **0.996** | 0.00013 | 0.976 | **0.994** | 0.00013 | 0.992 | **0.995** |

(ID: mnist)

| | OOD: cifar100 | | | OOD: tiny imagenet | | | OOD: mnist | | |
|---|---|---|---|---|---|---|---|---|---|
| Method | ↓ S. risk at TPR(0.80) FPR(0.21) | ↑ AUROC | ↑ OSCR | ↓ S. risk at TPR(0.80) FPR(0.19) | ↑ AUROC | ↑ OSCR | ↓ S. risk at TPR(0.80) FPR(0.19) | ↑ AUROC | ↑ OSCR |
| MSP [10] | 0.00676 | 0.871 | **0.977** | 0.00676 | 0.887 | **0.976** | 0.00676 | 0.899 | **0.976** |
| MLS [9] | 0.00984 | 0.861 | 0.973 | 0.00984 | 0.885 | 0.971 | 0.00984 | 0.905 | 0.971 |
| ODIN [11] | 0.01000 | 0.851 | 0.975 | 0.01000 | 0.864 | 0.974 | 0.00995 | 0.915 | 0.969 |
| REACT [17] | 0.00856 | 0.864 | 0.973 | 0.00856 | 0.888 | 0.971 | 0.00856 | 0.883 | 0.972 |
| KNN [19] | **0.00665** | **0.896** | 0.974 | **0.00665** | **0.914** | 0.972 | **0.00665** | **0.916** | 0.973 |
| VIM [20] | 0.01232 | 0.872 | 0.972 | 0.01232 | 0.888 | 0.971 | 0.01236 | 0.873 | 0.974 |
| KNN+MSP | **0.00652** | **0.896** | **0.977** | **0.00652** | 0.914 | 0.976 | **0.00652** | 0.916 | 0.976 |
| VIM+MSP | 0.00676 | 0.879 | **0.977** | 0.00676 | 0.894 | **0.976** | 0.00676 | 0.900 | **0.976** |

(ID: cifar10)

Table 2: Evaluation of existing single-score methods MSP, MLS, ODIN, REACT, KNN and two instances of the proposed double-score strategy: KNN+MSP and VIM+MSP. We use MNIST (top table) and CIFAR10 (bottom table) as ID, and three different datasets as OOD. We report the standard AUROC and OSCR, and the proposed selective risk at target TPR and FPR, where the selective risk corresponds to the classification error on accepted ID samples. Best results are in bold.

## 4 Experiments

In this section, we evaluate single-score OODD methods and the proposed double-score strategy, using the existing and the proposed evaluation metrics. We use MSP [10], MLS [9], ODIN [11] as baselines and REACT [17], KNN [19], VIM [20] as repesentatives of recent single-score approaches. We evaluate two instances of the double-score strategy. First, we combine the scores of MSP [10] and KNN [18] and, second, scores of MSP and VIM [20]. MSP score is asymptotically the best misclassification detector, while KNN and VIM are two best OOD/ID discriminators according to their AUROC. We always use the ID classifier of the MSP method. The evaluation data and implementations of OODD methods are taken from OpenOOD benchmark [21]. Because the datasets have unrealistically high portion of OOD samples, e.g., $\pi > 0.5$, we use metrics that do not depend on $\pi$. Namely, AUROC and OSCR as the most frequently used metrics, and the proposed selective risk at TPR and FPR. We use 0/1-loss, hence the reported selective risk is the classification error on accepted ID samples with guranteed TPR and FPR. In all experiments we fix the target TPR to 0.8 while FPR is set for each database to the highest FPR attained by all compared methods.

Results are presented in Tab. 2. It is seen that the single-score methods with the highest AUROC and OSCR are always different, which prevents us to create a single conclusive ranking of the evaluated approaches. MSP is almost consistently the best misclassification detector according to OSCR. The best OOD/ID discriminator is, according to AUROC, one of the recent methods: REACT, KNN, or VIM. The proposed double-score strategy, KNN+MSP and VIM+MSP, consistently outperforms the other approaches in all metrics.

## 5 Conclusions

This paper introduces novel reject option models which define the notion of the optimal prediction strategy for OOD setups. We prove that all models, despite their different formulations, share the same class of optimal prediction strategies. The main insight is that the optimal prediction strategy must trade-off the ability to detect misclassified examples and to distinguish ID from OOD samples. This is in contrast to existing OOD methods that output a single uncertainty score. We propose a simple and effective double-score strategy that allows us to boost performance of two existing OOD methods by combining their uncertainty scores. Finally, we suggest improved evaluation metrics for assessing OOD methods that simultaneously evaluate all aspects of the OOD methods and are directly related to the optimal OOD strategy under the proposed reject option models.

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
