# OpenReview forum: "Reject option models comprising out-of-distribution detection"
_NeurIPS.cc/2023/Conference — Submitted to NeurIPS 2023_

### Official Review · Reviewer_cqqv · 2023-07-04

**Soundness:** 2 fair
**Presentation:** 2 fair
**Contribution:** 2 fair
**Rating:** 4
**Confidence:** 3

**Summary:**

The authors consider 3 rejection models in the OOD detection setting and establish theorems about the optimal rules for these models. Due to the commonly used metrics that may partially only focus on either OOD detection or misclassification, they proposed a double-score method to consider both aspects.

**Strengths:**

* The authors extend the rejection in the non-OOD setting to the OOD detection setting.
* The 3 different rejection models share similar optimal strategies, and those strategies outperform other baselines.
* Since previous works only focus on either OOD detection or misclassification, the authors proposed a novel double-score method to consider both aspects.


**Weaknesses:**

* Now that we know all the distributions, including OOD and ID, why put effort into eq 3? Can’t we just treat augmented data of OOD + ID as the new collection with $K + 1$ "inlier" classes, then follow the regular rejection analysis under the closed-world assumption?

* The optimal rules heavily rely on the clear information (the distribution in the theorem or estimated from the sample) on OOD data. For example, the prior $\pi$ in the bounded precision-recall rejection model, the conditional probability (of OOD) in the bounded TPR-FPR rejection model, or even both in the cost-based model. However, it is challenging to access OOD data and estimate these probabilities in real-world applications within an open-world setting. This raises concerns about the practical guidance provided by these optimal rules.

* I have a reservation about the constraints in equation 8 or the setup of problem 1. In practice, it is more likely the threshold or the selective function is determined by only one constraint (even in your proof of theorem 2, there is nowhere about the constraint $\rho(c)$). For example, once the $\lambda$ is determined by one constraint, it does not necessarily satisfy another one, which is similar to the trade-off between the type I and type II errors or Neyman-Pearson classification. A similar argument applies to Problem 2.

* Optimal rules: the authors claim the optimal rules related to the Bayes classifier (lines 126, 170), but from theorems, it holds under the condition that the given $(h, c)$ is optimal. If we cannot find the optimal $h$, can we still say the Bayes classifier is optimal?


**Questions:**

* Line 137: typos in "Let $g(x)$ …. samples" $\rightarrow$ $p_O(x)/p_I(x)$... ratio of OOD and ID samples.

* Theorem 3: Do you mean any $h$ is optimal even if it is not the Bayes classifier? The optimality looks like is conditional on the given $h$, how could you guarantee the picked $h$ associated with the corresponding $c$ simultaneously achieve to the minimizer?

* Line 157: What does this notation mean: (TPR, $\phi$)?

* Eq 12: Since $h^*=h_B$, do you mean $r_B(x)$ instead of $r(x)$?

* Line 301 (FPR@TPRx, TNR@TPRx, CCR@FPRx): These metrics also control one metric @ percentage x and then compared to another one, depending on their definitions of positive/negative for the ID and OOD, which is similar to the idea of your proposed problem 1/2, like controlling two metrics (actually can only achieve one constraint in general) and compare the selective risk.

* Line 441: Could you explain why $\min_{q\in \mathcal{Y}}R_x(q)$ is in this form (are you saying the smallest risk is always achieved by making predictions even for OOD data?), while later you use $r(x)$ instead of $r_B(x)$ in $\min_{q\in \mathcal{Y}}R_x(q)$ (1st equality below line 443).

* Line 449: You miss the subscript $I$ under $p$.


**Limitations:**

See my concerns and questions above.

---

> ### Author Rebuttal · Authors · 2023-08-09
>
> ***Weaknesses:***
>
> R: “Now that we know all the distributions, including OOD and ID, why put effort into eq 3? Can’t we just treat augmented data of OOD + ID as the new collection with K+1 "inlier" classes, then follow the regular rejection analysis under the closed-world assumption?”
>
> A: We agree that the problem can be formulated as you suggest. The regular reject option setup, however, assumes a single cost for rejecting to predict, i.e. the model you propose would use the same cost for i) rejecting prediction on ID sample and ii) for rejecting OOD sample. In practice, however, the two decisions can have different consequences and hence it is more natural to allow different costs as we did in (3). Note that your setup is a special case of (3).
>
> R: “The optimal rules heavily rely on the clear information ...”
>
> A: We understand your concern about the assumption of knowing the OOD distribution in our paper. We would like to clarify that our intent is not to propose an algorithm reliant on this knowledge for practical use. Instead, we employ OOD distribution knowledge for precise problem formulation and deriving optimal strategies. The exact OOD distribution is unnecessary for practical use of our results. Firstly, the problem formulation facilitates reliable evaluation metrics for comparing OODD methods. While evaluating, OOD remains unknown, yet we possess a sample from both OOD and ID, such as CIFAR and MNIST. Our proposed metrics provide a distinct method comparison, addressing current inconsistencies in rankings and classifier performance assessment. Secondly, knowledge of the optimal solution's form narrows our classifier's hypothesis class. Specifically, we can search only in the space of double score strategies and we also know the desirable properties of the two scores. We demonstrate experimentally that searching in the space to double-score strategies on its own (without trying to invent any new scores) outperforms many existing methods which do not use the knowledge.
>
> R: “I have a reservation about the constraints in equation 8 or the setup of problem 1. ...”
>
> A: You are right that the second constraint in (8), i.e the upper bound on FPR $\rho(c)$, does not have to be active in some cases. However, in general it can be active, and it is necessary to specify the minimal requirements on the selection function as the ID/OOD detector. Please note that the second constraint is the only place in (8) which depends on OOD distribution. If you remove the second constraint, the problem (8) becomes the standard bounded-abstention model for non-OOD setup (please see the last paragraph on page 4).
>
> R: “Optimal rules: the authors claim the optimal rules related to the Bayes classifier (lines 126, 170), but from theorems, it holds under the condition that the given (h,c) is optimal. If we cannot find the optimal h, can we still say the Bayes classifier is optimal?”
>
> A: Theorem 2 asserts that the optimal pair (h,c) after replacing h by the Bayes classifier $h_B$ remains optimal, i.e. Theorem 2 justifies two-stage procedure to learn the optimal solution: i) first learn $h$, ii) learn $c$ for fixed $h$. Problem 1 defines the step ii) and Theorem 3 characterizes its optimal solution. It follows that the optimal selection function (9) has the same form regardless of which h was used, be it Bayes classifier or our best approximation of $h_B$.
>
> ***Questions:***
>
> R: “Line 137: typos in "Let g(x) …. samples" → pO(x)/pI(x)... ratio of OOD and ID samples.”
>
> A: Thank you.
>
> R: “Theorem 3: Do you mean any h is optimal even if it is not the Bayes classifier? ...”
>
> A: Theorem 3 characterizes the optimal selection function $c^*$ for a single a priory fixed h. Note that even if we pick bad h, we can still find the best selection function c for our choice, cf. Problem 1. Theorem 3 asserts that the form of the optimal selection function is the same for arbitrary h. Note however that a particular h has a different conditional risk $r(x)$ and therefore it has a different optimal selection function $c^*$.
>
> R: “Line 157: What does this notation mean: (TPR, ϕ)?”
>
> A: We use letter $\phi$ to denote the portion of accepted ID samples, defined by equ (6). In OOD setup, $\phi$ is usually called True Positive Rate (TPR). In non-OOD setup, the same quantity $\phi$ is usually called the coverage. In the paragraph you mention, we describe the relation to non-OOD setup, speaking about the coverage and hence we use “coverage (TPR,$\phi$)” to remind readers that these things are the same. We will add more verbal explanations.
>
> R: “Eq 12: Since h∗=hB, do you mean rB(x) instead of r(x)?”
>
> A: Yes, but in all the definitions of optimal strategies, see Thm 3,5, we explicitly state that $r(x)$ is the conditional risk of the used classifier h, i.e. it follows here that $h^*=h_B$. We will emphasize it.
>
> R: “Line 301 (FPR@TPRx, TNR@TPRx, CCR@FPRx): These metrics also control one metric @ percentage x and then compared to another one, depending on their definitions of positive/negative for the ID and OOD, which is similar to the idea of your proposed problem 1/2, like controlling two metrics (actually can only achieve one constraint in general) and compare the selective risk.”
>
> A: Yes, that is the point. These metrics control only one quantity and report another one. However, in non-OOD setup we need to bind three quantities (metrics) at the same time.
>
> R: ”Line 441: Could you explain why minq∈YRx(q) is in this form (are you saying the smallest risk is always achieved by making predictions even for OOD data?), while later you use r(x) instead of rB(x) in minq∈YRx(q) (1st equality below line 443).”
>
> A: In equ 443, 1st equality, there should be $r_B(x) $instead of r(x). Thanks for pointing this.
>
> R: “Line 449: You miss the subscript I under p.”
>
> A: Yes, thanks.

---

> > ### Comment · Reviewer_cqqv · 2023-08-17
> >
> > Thanks for the author’s explanations.
> >
> > > In practice, however, the two decisions can have different consequences...
> >
> > * I fully agree with your observation that setting equal costs for the rejections of ID and OOD samples might not capture the potential different consequences associated with these two types of rejections. However, even with the introduction of distinct costs in the problem formulation, the proposed double-score rule, which forms the basis of the final decision, does not either differentiate between these ID and OOD rejections.
> >
> > > "The goal of the experiments is to prove that the theoretical concepts proposed in the paper are valid and can be readily applied in practice". "straightforward application of the double-score concept yields superior performance"
> >
> > * I concur that this article offers some conceptual problems, however, I don’t think so far it can be applied in practice since it does not solve the issue of access to OOD points. Without the usage of clear OOD data or its legitimate estimation, it is overly confident to claim the method renders the reliability in practice when it comes to the topic of “Safety in Machine Learning”.
> >
> > * By the same token, while the theoretical merits of the double-score concept are evident, its applicability is limited in practice by its dependence on the OOD distribution. In terms of the component of the proposed double-score, I think the discussion on the extra hyper-parameter $\mu$ is insufficient. It looks like that $\mu$ depends on how much the researcher trade-off among classification, ID, and OOD rejection. Is there any guidance to determine the value of $\mu$? Without clear discussion or even a rule, the decision-making process diminishes overall trustworthiness. In summary, I feel this article at least cannot bypass the limited contribution when deployed in reality. By the way, I just find the notation $\mu$ is abused between its original meaning and the mean value of Gaussian distribution (line 217).
> >
> > > ... in general it can be active ...
> >
> > * One more notable apprehension is still about the two formulated constrained problems (the authors' previous response cannot relieve my concern). Particularly, I do not believe “in general” two constraints can be simultaneously fulfilled. Once $\lambda$ is adjusted to satisfy the first constraint, $c(x)$ becomes fixed. However, this fixed $c(x)$, when plugged back into the equation (7), raises doubts about how it can “in general” lead to a false positive rate (FPR) lower than the pre-specified tolerance $\rho_{\text{max}}$ in equation (8). Achieving a high true positive rate (TPR) necessitates sacrificing the FPR, an intuitive trade-off.

---

> > > ### Author Response · Authors · 2023-08-18
> > >
> > > > ... However, even with the introduction of distinct costs in the problem formulation, the proposed double-score rule, which forms the basis of the final decision, does not either differentiate between these ID and OOD rejections.
> > >
> > > We respectfully disagree. The optimal decision rule for cost-based formulation is outlined in (5), depending on the costs for rejecting ID sample $\epsilon_1$ and non-rejecting OOD sample $\epsilon_2$.
> > >
> > > > I concur that this article offers some conceptual problems, however, I don’t think so far it can be applied in practice since it does not solve the issue of access to OOD points. Without the usage of clear OOD data or its legitimate estimation, it is overly confident to claim the method renders the reliability in practice when it comes to the topic of "Safety in Machine Learning".
> > >
> > > Existing single-score OODD methods, including those applied in practice, require tuning the decision threshold $\lambda$ using a small validation sample containing clear OOD data. In our proposed double-score approach, tuning involves $\lambda$ and an additional parameter $\mu$, which we believe is not a significant practical issue. It's important to note that the main challenge lies in learning the score $g(x)=p_O(x)/p_I(x)$ due to limited OOD information, which we do not address in the paper. The posthoc tuning of one (for existing methods) or two (in our case) parameters is a secondary concern.
> > >
> > > > By the same token, while the theoretical merits of the double-score concept are evident, its applicability is limited in practice by its dependence on the OOD distribution. In terms of the component of the proposed double-score, I think the discussion on the extra hyper-parameter  is insufficient. It looks like that  depends on how much the researcher trade-off among classification, ID, and OOD rejection. Is there any guidance to determine the value of ? Without clear discussion or even a rule, the decision-making process diminishes overall trustworthiness. In summary, I feel this article at least cannot bypass the limited contribution when deployed in reality. By the way, I just find the notation  is abused between its original meaning and the mean value of Gaussian distribution (line 217).
> > >
> > > Clear instructions are provided. To determine the parameters $\mu$ and $\lambda$, solve equation (13) for Bounded TPR-FPR formulation, or equation (14) for Bounded Prec-Recall formulation. Please note that the rule set $Q$ optimized in (13) and (14) is parametrized by $\mu$ and $\lambda$. The algorithm to solve (13) and (14) is explained in Appendix B.2 due to space limitations.
> > > > ... By the way, I just find the notation  is abused between its original meaning and the mean value of Gaussian distribution (line 217).
> > >
> > > Indeed, this is a mistake. Thanks. We will fix it.
> > >
> > > > One more notable apprehension is still about the two formulated constrained problems (the authors' previous response cannot relieve my concern). Particularly, I do not believe “in general” two constraints can be simultaneously fulfilled. Once λ is adjusted to satisfy the first constraint, c(x) becomes fixed. However, this fixed c(x), when plugged back into the equation (7), raises doubts about how it can “in general” lead to a false positive rate (FPR) lower than the pre-specified tolerance ρmax in equation (8). Achieving a high true positive rate (TPR) necessitates sacrificing the FPR, an intuitive trade-off.
> > >
> > > The confusion arises from the misconception that we fix $\lambda$ first and then adjust $\mu$. It's important to clarify that we actually adjust $\lambda$ and $\mu$ simultaneously when solving (13) and (14), respectively. Refer to the previous comment for further context.
> > >
> > > We appreciate your observations regarding the potential ambiguity in comprehending the paper. To address this, we will include a clarifying note that $\mu$ and $\lambda$ are tuned using the approaches defined in problems (13) and (14).

---

### Official Review · Reviewer_msSt · 2023-07-06

**Soundness:** 3 good
**Presentation:** 2 fair
**Contribution:** 3 good
**Rating:** 4
**Confidence:** 4

**Summary:**

This paper provides a unified viewpoint of reject model evaluation metrics under OOD-ness. Despite different performance metrics being used in practice, they can all be factorized into both the misclassification prediction on ID datapoints as well as the discrimination performance between ID and OOD. The experimental section provides some evidence that this proposed two score decomposition can provide competitive OOD detection and input rejection performance.

**Strengths:**

- OOD detection and classifying with a reject option are both timely and important issues in trustworthy machine learning. ML algorithms need to be aware of their limitations and signal if they are deployed on previously unseen data.
- The unified framework / viewpoint makes sense and yields an interesting decomposition result showing that the optimal uncertainty score is a function of both misclassification prediction on ID datapoints as well as distinction performance between ID and OOD. Although this appears like an intuitive desiderata for a prediction uncertainty score, explicitly tying it back to the different rejection models seems new.

**Weaknesses:**

- The paper assumes that the OOD distribution is known. This is a strong assumption and severely limits the applicability of the approach in practical settings where knowing or explicitly estimating this distribution is often prohibitive.
- Even though the paper does provide a nice unification of three performance measures for reject option models in an OOD setup (cost-based rejection, TPR-FPR rejection, precision-recall rejection), these metrics are not new and routinely used in existing applications. To me it seems like this kind of unified viewpoint would be better suited for a survey-style paper.
- Section 3 could have been motivated better. To me, it appears a bit sudden without much justification or transition from the previous section.
- Section 3.1 could have significantly profited from a figure showcasing the 1D Gaussian example for added intuition.
- While the optimal uncertainty score is unified across error models, the experimental section showing the efficacy of the double score method is very lackluster. Details about models, training methods, and hyper-parameters area all missing (and also not documented in the appendix). Results are also based on a single run which makes it impossible to judge the statistical significance of the presented results. Moreover, the considered datasets do not all share the same sample shapes (e.g. MNIST vs CIFAR-10) which makes me wonder how OOD scores were obtained here in the first place. Real life shifts, like from the WILDS dataset collection would have been a better place to assess OOD-ness of samples. I am not convinced by this evaluation.

### Post-rebuttal

I have read the author's rebuttal and found that although many of my concerns were adequately addressed, the experimental documentation issue remained unaddressed. I increased my score only slightly as a result.

**Questions:**

- Can you add additional motivation for Section 3?
- Can you provide more experimental details?

**Limitations:**

Weaknesses of the approach are not adequately discussed in the paper. See weaknesses and questions above.

---

> ### Author Rebuttal · Authors · 2023-08-09
>
> ***Weaknesses:***
>
> R: “The paper assumes that the OOD distribution is known. This is a strong assumption and severely limits the applicability of the approach in practical settings where knowing or explicitly estimating this distribution is often prohibitive.”
>
> A: Thank you for your valuable feedback. Regarding the assumption about knowing the OOD distribution, please note that our intent is not to propose an algorithm reliant on this knowledge for practical use. Instead, we employ OOD distribution knowledge for precise problem formulation and deriving optimal strategies. The exact OOD distribution is unnecessary for practical use of our results. Firstly, the problem formulation facilitates reliable evaluation metrics for comparing OODD methods. While evaluating, OOD remains unknown, yet we possess a sample from both OOD and ID, such as CIFAR and MNIST. Our proposed metrics provide a distinct method comparison, addressing current inconsistencies in rankings and classifier performance assessment. Secondly, knowledge of the optimal solution's form narrows our classifier's hypothesis class. Specifically, we can search only in the space of double score strategies and we also know the desirable properties of the two scores. We demonstrate experimentally that searching in the space to double-score strategies on its own (without trying to invent any new scores) outperforms many existing methods which do not use the knowledge.
>
> R: “Even though the paper does provide a nice unification of three performance measures for reject option models in an OOD setup (cost-based rejection, TPR-FPR rejection, precision-recall rejection), these metrics are not new and routinely used in existing applications. To me it seems like this kind of unified viewpoint would be better suited for a survey-style paper.”
>
> A: Please notice a difference between the reject option model and the evaluation metric (performance measure). Note that we i) propose three formulations of reject option models, Def. 1, 2, 3 (reject option model = optimization problems where the variable is a decision strategy) and ii) we derive evaluation metrics for two cost-less reject option models from Def 2 and 3 (evaluation metric = a scalar function whose inputs are test/validation samples and parameters of the reject option model, e.g. target TPR and FPR). To your comment: First, we are not aware of any paper exactly formulating models in Def. 1,2,3. We would like to ask for a reference to prove that we are wrong. Second, the existing papers use evaluation metrics that measure the performance of the classifier and the selection function in isolation and we argue that it is not enough. In contrast, our evaluation metrics take into account the performance of both components simultaneously which we believe to be a novel approach. Please see the discussion in Sec 3.4.
>
> R: “Section 3 could have been motivated better. To me, it appears a bit sudden without much justification or transition from the previous section.”
>
> A: We will try to motivate it better. Please see the answer below.
>
> R: “Section 3.1 could have significantly profited from a figure showcasing the 1D Gaussian example for added intuition.”
>
> A: We fully agree. Please see Figure 1 in the appendix which is not in the main paper due to lack of space.
>
> R: “While the optimal uncertainty score is unified across error models, the experimental section showing the efficacy of the double score method is very lackluster. Details about models, training methods, and hyper-parameters are all missing (and also not documented in the appendix). Results are also based on a single run which makes it impossible to judge the statistical significance of the presented results. Moreover, the considered datasets do not all share the same sample shapes (e.g. MNIST vs CIFAR-10) which makes me wonder how OOD scores were obtained here in the first place. Real-life shifts, like from the WILDS dataset collection would have been a better place to assess OOD-ness of samples. I am not convinced by this evaluation.”
>
> A: The reviewer is correct that a detailed description of the experiments is missing from the paper. This is due to a lack of space. However, note that i) the experiments use trained models, scoring functions, evaluation protocol, and data published in [21] and ii) we provide full code in the appendix.
>
> ***Questions:***
>
> R: “Can you add additional motivation for Section 3?”
>
> A: The exact formulation of the reject option models for OOD setup, which are described in Sec 2, allows us to design novel evaluation metrics that provide a reliable assessment of OODD methods. In Sec 3, we propose such evaluation metrics, we explain them on synthetic data and we compare them against the existing metrics used so far.
>
> R: “Can you provide more experimental details?”
>
> A: Implementation of the compared methods, pre-trained models, the data, and their splits into training/validation/test sets are all adopted from the OpenOOD [21]. We use the OpenOOD benchmark to obtain the tuple (true label, predicted label, score, ID/OOD) for each sample. Only the proposed evaluation metrics (described in Sec 3) and the double-score methods (described in Sec 2.4) are new. We also submitted the working code (cf. supplementary) which can be used to replicate the results and is planned to be made public. We will emphasize it in Sec 4.

---

> > ### Comment · Reviewer_msSt · 2023-08-10
> > **Rebuttal Response**
> >
> > I thank the authors for their response to my comments.
> >
> > > A: Thank you for your valuable feedback. Regarding the assumption about knowing the OOD distribution [...]
> >
> > Thanks for clarifying. It would be important to highlight the fact that practical applications of the concepts from the paper do not require explicit knowledge of the OOD distribution. If I understand the response correctly, then sample access to the OOD distribution is still required. That is a weaker assumption but still might be restrictive in many practical scenarios.
> >
> > > A: Please notice a difference between the reject option model and the evaluation metric [...]
> >
> > > A: The exact formulation of the reject option models for OOD setup, which are described in Sec 2 [...]
> >
> > Thanks for elaborating on this. In combination with the additional elaboration on the purpose of Section 3. This is more clear to me now.
> >
> > > A: We fully agree. Please see Figure 1 in the appendix which is not in the main paper due to lack of space.
> >
> > Thanks for the pointer. It would have been great if the data was visualized. Since the data is 1D this should have been simple (and could have been included in the rebuttal PDF).
> >
> > > A: The reviewer is correct that a detailed description of the experiments is missing from the paper. This is due to a lack of space. However, note that i) the experiments use trained models, scoring functions, evaluation protocol, and data published in [21] and ii) we provide full code in the appendix.
> >
> > > A: Implementation of the compared methods, pre-trained models, the data, and their splits into training/validation/test sets are all adopted from the OpenOOD [21].
> >
> > Unfortunately, I am not satisfied with these responses. At the very least, the appendix should have detailed the experimental setup, even if it is mostly shared from previous works. Moreover, my comment on MNIST vs CIFAR-10 as well as the proposal to consider more real-life shifts was not addressed by the authors.
> >
> > ---
> > I have increased my score in light of the clarification around section 3. I still have some reservations wrt the OOD distribution assumption and especially the experimental evaluation which prevents me from further raising my score.

---

> > > ### Author Response · Authors · 2023-08-15
> > >
> > > We are glad that our response clarified most of your questions.
> > >
> > > We see that you still have a reservation regarding the need of OOD samples. We would like to emphasize that we require OOD samples only for the final evaluation of the compared methods. Please note that empirical evaluation without having samples drawn from the underlying true distribution, which in our case is a mixture of ID and OOD distributions, is NOT POSSIBLE. Please note that this assumption is not specific for our OOD setup, but it is used in all existing papers.
> > >
> > > We thank you again for your valuable time spent on our paper and for raising your score.

---

> > > > ### Comment · Reviewer_msSt · 2023-08-17
> > > > **Thank you**
> > > >
> > > > Thank you for additionally clarifying the OOD sample access assumption.
> > > >
> > > > Unfortunately, my last concern on the experimental setup is still left unaddressed by the authors which prevents me from further raising my score at the moment.

---

> > > > > ### Author Response · Authors · 2023-08-17
> > > > >
> > > > > OpenOOD [21] offers a streamlined process to assess OODD methods. With a single script call, it provides uncertainty scores, predicted and ground-truth labels for an OODD method, training, and test dataset triplet. This data enables evaluation using established or novel metrics. For the double-score strategy, we combine scores from two chosen OODD methods as outlined in our paper.
> > > > >
> > > > > This description will be added to the revised paper. Please inform us of any unclear aspects in the experimental setup. We will be happy to answer it.
> > > > >
> > > > > As previously stated, while we recognize the potential value of comprehensive empirical evaluations on larger datasets, our paper's main emphasis lies elsewhere. Thank you once more for your time.

---

### Official Review · Reviewer_77aK · 2023-07-07

**Soundness:** 3 good
**Presentation:** 3 good
**Contribution:** 3 good
**Rating:** 7
**Confidence:** 3

**Summary:**

This paper proposes three reject option models and introduces double-score OOD methods that consistently outperform state-of-the-art methods. The authors also propose novel evaluation metrics for comprehensive and reliable assessment of OOD methods. The proposed metrics simultaneously evaluate the classification performance on the accepted ID samples and guarantee the performance of the OOD/ID discriminator, either via constraints in TPR-FPR or Precision-Recall pair. The authors argue that setting these extra parameters is better than using the existing metrics that provide incomplete if used separately, or inconsistent, if used in combination, view of the evaluated methods.  Overall, the paper's contributions provide a significant improvement in OOD detection and evaluation.

**Strengths:**

The paper proposes three reject option models for OOD setups, which extend the standard reject option models. These models define the notion of an optimal OOD selective classifier and establish that all the proposed models, despite their different formulations, share a common class of optimal strategies. This is an original and creative approach to the problem of OOD detection, and the proposed models are well-motivated and clearly explained. The paper introduces double-score OOD methods that leverage uncertainty scores from two chosen OOD detectors: one focused on OOD/ID discrimination and the other on misclassification detection. The paper proposes novel evaluation metrics derived from the definition of the optimal strategy under the proposed OOD rejection models. These metrics provide a comprehensive and reliable assessment of OOD methods.

Overall, the paper is well-written and easy to follow, with clear explanations of the proposed models, methods, and evaluation metrics.

**Weaknesses:**

One weakness of the paper in the experimental results is that the dataset used in the experiments is relatively small, and the proposed methods do not show significant advantages over the baseline. This raises questions about the generalizability of the proposed methods to larger and more diverse datasets. Additionally, the paper could benefit from a more detailed analysis of the limitations of the proposed methods and how they can be improved in future work.

**Questions:**

1. The dataset used in the experiments of the paper is relatively small, and the proposed methods do not show significant advantages over the baseline in terms of experimental results. It would be helpful for the quality of the paper if the authors could provide more compelling experimental results or demonstrate the effectiveness of the proposed methods on larger datasets.

**Limitations:**

The authors did not explicitly discuss the limitation of the proposed method.

---

> ### Author Rebuttal · Authors · 2023-08-09
>
> ***Weaknesses:***
>
> R: “…Additionally, the paper could benefit from a more detailed analysis of the limitations of the proposed methods and how they can be improved in future work.”
>
> A: We discuss the main limitations in Sec 3.5. We fully agree that the discussion could be more detailed and that future work is not covered. Such extension, however, would require either to remove some other content (which?) or to further compress the text which is already concise.
>
> ***Questions:***
>
> R: “The dataset used in the experiments of the paper is relatively small, and the proposed methods do not show significant advantages over the baseline in terms of experimental results. It would be helpful for the quality of the paper if the authors could provide more compelling experimental results or demonstrate the effectiveness of the proposed methods on larger datasets.”
>
> A: We agree that large empirical evaluation using larger datasets would be interesting, however, it is not the ambition of our paper. Our paper provides a new reliable methodology for such a comparison. The goal of the experiments is to prove that the theoretical concepts proposed in the paper are valid and can be readily applied in practice. Namely, we want to i) show that the proposed evaluation metrics provide a different view on the compared methods than the existing metrics which have their limitations and ii) demonstrate that even straightforward use of the concept of double-score methods leads to superior performance in terms of the proposed metrics.

---

> > ### Comment · Reviewer_77aK · 2023-08-16
> >
> > I thank the author for the response. I am still inclined to give this paper a positive rating. I believe that additional experimental validation and analysis would further enhance the impact of the paper.

---

> > > ### Author Response · Authors · 2023-08-16
> > >
> > > We thank you for your time spent on our paper and for your positive rating.

---

### Official Review · Reviewer_G4Mq · 2023-07-07

**Soundness:** 3 good
**Presentation:** 3 good
**Contribution:** 2 fair
**Rating:** 5
**Confidence:** 4

**Summary:**

This paper addresses out-of-distribution detection (ID/OOD discrimination) and misclassification detection (selection classification). The authors introduce double-score OOD methods that leverage uncertainty scores from OOD detector and misclassification detector. For evaluation metric, this paper proposes to use PR curve.

**Strengths:**

(1) Existing works mainly focus on OOD detection, which this paper simultaneously considers OOD and selection classification. This is more practical and useful for high risk applications.

(2) Theoretical analysis of OOD detection and selective classification is valuable.

**Weaknesses:**

(1) The analysis of Bayes-optimal OOD selective classifier as well as the Bayes-optimal misclassification selective classifier has also been investigated in [1]. The authors are suggested to demonstrate the difference.

(2) The experiments are insufficient, only evaluating on cifar-10 and mnist. Results on CIFAR-100 is valuable.

(3) What is the advantage of PR curve over AURC (risk-coverage) curve [2] for evaluating rejection model?

[1] Narasimhan, H., Menon, A. K., Jitkrittum, W., & Kumar, S. (2023). Learning to reject meets OOD detection: Are all abstentions created equal?. arXiv preprint arXiv:2301.12386.

[2] Geifman, Y., & El-Yaniv, R. (2017). Selective classification for deep neural networks. Advances in neural information processing systems, 30.

**Questions:**

A recent work [3] also addresses the selective classification and OOD detection at the same time. it is worth mentioning it.

[3] Zhu, F., Cheng, Z., Zhang, X. Y., & Liu, C. L. (2023). OpenMix: Exploring Outlier Samples for Misclassification Detection. In Proceedings of the IEEE/CVF Conference on Computer Vision and Pattern Recognition (pp. 12074-12083).

---

> ### Author Rebuttal · Authors · 2023-08-09
>
> ***Weaknesses:***
>
> R: “ (1) The analysis of Bayes-optimal OOD selective classifier as well as the Bayes-optimal misclassification selective classifier has also been investigated in [1]. The authors are suggested to demonstrate the difference.
>
> [1] Narasimhan, H., Menon, A. K., Jitkrittum, W., & Kumar, S. (2023). Learning to reject meets OOD detection: Are all abstentions created equal?. arXiv preprint arXiv:2301.12386.”
>
> A: We agree with the reviewer that [1] is relevant and very close to our paper. However note that [1] is i) un-reviewed work and ii) it was published after NeurIPS submission deadline, in July 2023. Therefore the paper was not public as of the time of submission and we are therefore not sure whether the work should be addressed in our paper. Nevertheless, the authors of [1] also formulate and provide an optimal solution of a reject option model for OOD setup which is however different from our models. Their model relies on defining a cost for accepting OOD samples. We try to avoid such formulation, because costs of misclassification and accepting OOD samples have different units and are different to balance. We instead propose the cost-less models, i.e. Bounded TPR-FPR and bounded Prec-Recall model, which allow specifying the desired properties of the model. The authors [1] also show that the optimal solution of their model leads to double-score strategy. However, they implicitly assume continuous input spaces and loss functions while our characterization of the optimal strategy is fully general and valid for any probability distribution and any loss. E.g. they do not reveal that in the discrete case the optimal strategy needs to randomize in the corner cases, etc.
>
> R: “(2) The experiments are insufficient, only evaluating on cifar-10 and mnist. Results on CIFAR-100 is valuable.”
>
> A: We agree that large empirical evaluation using more OODD methods and datasets would be interesting, however, it is not the ambition of our paper. Our paper provides a new reliable methodology for such a comparison. The goal of the experiments is to prove that the theoretical concepts proposed in the paper are valid and can be readily applied in practice. Namely, we want to i) show that the proposed evaluation metrics provide a different view on the compared methods than the existing metrics and ii) demonstrate that even straightforward use of the concept of double-score methods leads to superior performance in terms of the proposed metrics.
>
> R: “(3) What is the advantage of PR curve over AURC (risk-coverage) curve [2] for evaluating rejection model?
>
> [2] Geifman, Y., & El-Yaniv, R. (2017). Selective classification for deep neural networks. Advances in neural information processing systems, 30.”
>
> A: Please note that we do not propose to use the Precision-Recall (PR) curve to evaluate the rejection model. We propose to use the PR curve just to specify the admissible values of the precision and recall, however, the main metric is still the selective risk. Once the target precision and recall are selected (reflecting the application needs), the model should be evaluated by reporting the selective risk at the given precision and recall (please see Sec 3.3). Regarding the AURC (area under Risk-Coverage curve): AURC in isolation does not capture the performance of the model when OOD samples are present. It is used to evaluate the performance of the selective classifier in the standard non-OOD setup [2]. So, the answer to your question is: the advantage of using i) PR curve and selective risk instead of ii) AURC, is that AURC evaluates only performance of the selective classifier on the ID data. However, selective risk at a given PR evaluates both, performance of the classifier on ID data and the ability of the model to distinguish ID from OOD. Please see Sec 3.4. for a more detailed explanation.
>
> ***Questions:***
>
> R: “A recent work [3] also addresses the selective classification and OOD detection at the same time. it is worth mentioning it.
>
> [3] Zhu, F., Cheng, Z., Zhang, X. Y., & Liu, C. L. (2023). OpenMix: Exploring Outlier Samples for Misclassification Detection. In Proceedings of the IEEE/CVF Conference on Computer Vision and Pattern Recognition (pp. 12074-12083).”
>
> A: Thanks for the pointer. [3] proposes to leverage OOD samples to improve the performance of the misclassification detector of ID samples. Their primal goal is to train reject option classifier (they do not use the decomposition to the classifier and selection function) in standard non-OOD setup, i.e. they do not consider OOD samples in the test distribution as we do. Nevertheless in their analysis of the problem they write on page 6: “This observation points out an interesting future research direction of developing confidence estimation methods that consider OOD detection and MisD in a unified manner.”, which is exactly what we are paper attempts to do.

---

### Official Review · Reviewer_hjaD · 2023-07-17

**Soundness:** 4 excellent
**Presentation:** 3 good
**Contribution:** 3 good
**Rating:** 6
**Confidence:** 3

**Summary:**

The paper presents a formal analysis of three distinct models for classifiers with a reject option in the presence of out-of-distribution inputs at test time. All three models, viz. 1) cost-based rejection, 2) bounded TPR-FPR, and 3) bounded precision-recall, share the same form of optimal selective classifier (see Section 2.4).

This selective classifier consists of the Bayes in-distribution classifier $h_B(x)$ (which is optimal for a given in-distribution), and a selection function score that is a linear combination of the conditional risk $r(x)$ and the likelihood-ratio of OOD to ID $g(x)$. Based on this analysis, the authors point out the limitations of current OOD detection methods which only have a single score, and instead propose double-score OOD detection methods which can focus on both mis-classification detection and ID/OOD separation.

Using a concrete synthetic example, they discuss the limitations of existing metrics such as AUROC and AUPR in evaluating selective classifiers in the OOD setting. They propose a novel metric (one each for the bounded TPR-FPR and bounded precision-recall models) which calculates the selective risk, subject to a given minimum TPR and maximum FPR (or a given minimum recall and minimum precision). The proposed metric is shown to be better at capturing the overall performance of selective classifiers in the OOD setting.

**Strengths:**

1. The proposed reject option models and evaluation metrics are developed systematically and connected well with summary sections. Found the paper to be well written.

2. The analysis builds upon the prior work [6] which deals with reject option models but not in the OOD setting. In this work, they consider rejecting inputs due to both mis-classification and being OOD.

3. Although the proposed work makes a strong assumption of a known OOD distribution, the analysis is useful to draw insights about the need for a double-score OOD detection method, and also to highlight the shortcomings of existing evaluation metrics.

[6] https://www.jmlr.org/papers/volume24/21-0048/21-0048.pdf

**Weaknesses:**

1. The analysis and proposed new metrics (parts of the new metrics like the FPR and precision) depend upon knowledge of the OOD distribution. This is a strong assumption for practical settings.

2. For the proposed double-score OODD method, it seems to me that we need access to OOD data in order to set the hyper-parameter $\mu$ in the combined score $s_r(x) + \mu s_g(x)$. The authors should clarify if this is set based on validation data from a different OOD distribution than the test data.

3. Minor: it is a bit tedious to keep track of all the notations needed for the analysis.

**Update after author-reviewer discussions:** \
I have read the authors rebuttal. The paper lacks sufficient discussion on the experiments. Not enough details are provided in the appendix as well. Hence, I decrease my rating to 6.

**Questions:**

1. What is the metric OSCR on line 37? This is defined much later in the paper, please add a reference earlier.

2. Suggestion: “Selection function” seems like a better term than “Selective function”.

3. Line 28: “rejects it to predict” → “rejects it from prediction”.

4. Line 87: in the definition of the expected risk, the expectation should be over $(x, \bar{y}) \sim p(x, \bar{y})$.

5. Some notations are not defined, e.g, the indicator function $c_C(x) = [[ s_C(x) \leq \epsilon_1 ]]$ on line 96 (likewise on line 145). The logical AND symbol $\wedge$ in Eqn (3).

6. In the equation for $R^S(h, c)$ on line 117, the arguments of the loss function seem to be swapped. Should be $\ell(y, h(x))$.

7. In equation (8), it’s not clear to me what the sets $\mathcal{Y}^{\mathcal{X}}$ and $[0, 1]^{\mathcal{X}}$ mean. Please clarify.

8. Statement of Theorem 3 has a minor typo: should be $g(x) = p_O(x) / p_I(x)$.

9. The validation set $\mathcal{T}$ at the start of Section 3 is assumed to have OOD samples from the unknown OOD distribution $p_O(x)$. The OOD samples from the test set follow this same distribution. Is this a reasonable assumption to make for post-hoc tuning? Would not be true in most practical scenarios.

10. Typo on line 230: should be $h^*(x)$.

11. Lines 257, 258: the method should be $D(\mathbb{R})$ and not $E(\mathbb{R})$.

12. For the proposed double-score OODD method denoted by $D(\mathbb{R})$, how is the constant $\mu$ set (e.g. in Table 1)? Is it set by searching for the value that leads to the smallest selective risk (in this case selective error rate) subject to the TPR/FPR constraints? If this is the case, then it could be considered unfair since the OOD distribution in the test dataset is very likely to be different from that of the validation dataset $\mathcal{T}$. Could you clarify this point?

13. In Table 1, why are the AUROC values the same for the methods $A(\infty)$ and $D(\mathbb{R})$? Same question for the AUPR metric.

14. For the experiments in Table 2, can you explain how the methods KNN+MSP and VIM+MSP are combined for the proposed double-score method? As described on lines 202 - 203, one of the OOD methods is used for OOD/ID discrimination and other used for misclassification detection. How do you set the constant $\mu$ used for combining the scores?

15. In Table 2, why is the selective risk at given TPR and FPR the same (or almost the same) for multiple OOD datasets? For example, for CIFAR-10 as in-distribution, all the methods (including the proposed KNN+MSP and VIM+MSP) have the same selective risk across all three OOD datasets.

**Limitations:**

Some limitations of the proposed work are mentioned in Section 3.5. Another limitation to include is the fact that the proposed analysis and novel metrics depend on the OOD distribution which is usually unknown. One could use a validation set (as in the paper) with a mix of in-distribution and auxiliary OOD data, but the distribution of auxiliary OOD data in the validation and test sets should be different.

Negative societal impacts has not been discussed, but may not be applicable here.

---

> ### Author Rebuttal · Authors · 2023-08-09
>
> ***Weaknesses:***
>
> R: “The analysis and proposed new metrics (parts of the new metrics like the FPR and precision) depend upon knowledge of the OOD distribution. This is a strong assumption for practical settings.”
>
> A: Thank you for your valuable feedback. We understand your concern about the assumption of knowing the OOD distribution. We would like to clarify that we leverage OOD distribution knowledge only for precise formulation of the problem and deriving optimal strategies. Practical use doesn't demand exact OOD knowledge. Firstly, the problem formulation allows to design reliable evaluation metrics to assess and compare OODD methods. Note that when evaluating the methods, OOD is unknown but we have a sample drawn from OOD and ID, e.g. CIFAR is ID and MNIST is OOD sample. Secondly, understanding the optimal solution's structure narrows our classifier's hypothesis class. As we showed experimentally, this targeted search, focusing on double-score strategies with known properties, surpasses many existing methods without introducing new scores.
>
> R: “For the proposed double-score OODD method, it seems to me that we need access to OOD data in order to set the hyper-parameter  in the combined score . The authors should clarify if this is set based on validation data from a different OOD distribution than the test data.”
>
> A: You are right that to set the hyper-parameters we need sample of OOD data. Please note that all existing papers, proposing single-score methods, have exactly the same requirement as they need to tune the decision threshold. Our method requires to setup the decision threshold and, in addition, $\mu$ balancing the two scores, i.e. instead of a single hyper-parameter we tune two hyper-parameters. Therefore, in either case, the validation set can be quite small because the chance of over-fitting the selective function (which is essentially two-class classifier in 2D) is low.
>
> ***Questions:***
>
> R: “1. What is the metric OSCR on line 37? This is defined much later in the paper, please add a reference earlier.”
>
> A: Yes.
>
> R: “2. Suggestion: “Selection function” seems ...”
>
> A: Yes, we will use “selection function”.
>
> R: “3. Line 28: “rejects it to predict” → “rejects it from prediction”.”
>
> A: Yes.
>
> R: “4. Line 87: in the definition of the expected ...”
>
> A: Yes.
>
> R: “5. Some notations are not defined ...”
>
> A: We will define the Iverson bracket. We think that defining the logical AND symbol $\land$ is not necessary.
>
> R: “6. In the equation for RS(h,c) on line 117...”
>
> A: Yes, however, most classification losses are symmetric. Anyway, we will swap it as you suggest.
>
> R: “7. In equation (8), it’s not clear to me what the sets YX and [0,1]X mean. Please clarify.”
>
> A: $[0,1]^X$ is a common notation for a space of all functions from $X$ to $[0,1]$. Similarly, $Y^X$ denotes a space of functions from $X$ to $Y$.
>
> R: “8. Statement of Theorem 3 has a minor typo: should be g(x)=pO(x)/pI(x).”
>
> A: Yes.
>
> R: “9. The validation set T at the start of Section 3 is assumed ...”
>
> A: The assumption is that the validation set contains samples from the ID and OOD. All papers we are aware of use the very same assumption. The assumption is essential and cannot be avoided in order to estimate the performance on the test set (unless you impose an assumption about the ID and OOD models). Note however that our metrics used in the experiments do not depend on the proportion of ID and OOD, i.e. it does not have to be captured by the validation set.
>
> R: “10. Typo on line 230: should be h∗(x).”
>
> A: Yes.
>
> R: “11. Lines 257, 258: the method should be D(R) and not E(R).”
>
> A: Yes.
>
> R: “12. For the proposed double-score OODD method denoted ...”
>
> A: $\mu$ is found by solving the problem (13) on validation data. We assume that the validation data represent the test distribution. Please see the answer to question 9.
>
> R: “13. In Table 1, why are the AUROC values the same for the methods A(∞) and D(R)? Same question for the AUPR metric.”
>
> A: The metrics in Table 1 are reported for the best score from the hypothesis space Q produced by the particular method. The method $A(\infty)$ by construction uses Q containing the optimal OOD/ID discriminators, hence it achieves the best AUROC and AUPR (i.e. metrics capturing performance of the OOD/ID discriminators). The method $C(0)$ uses Q containing the optimal misclassification detectors, hence it achieves the best OSCR (i.e. metric capturing the performance of the misclassification detectors). The proposed method $D(\Re)$ by construction uses the hypotheses space $Q$ containing the hypothesis spaces of methods $A(\infty)$ and $C(0)$, i.e. it must attain the best value w.r.t any metric used (AUROC/OSCR/AUPR).Therefore AUROC/AUPR of $A(\infty)$ and $D(\Re)$ are equal, similarly, OSCR of $C(0)$ and $D(\Re)$ are equal.
>
> R: “14. For the experiments in Table 2, can you explain ...”
>
> A: All compared methods are post-hoc approaches which output the same classifier h and different uncertainty score s. Let $s_{KNN}$ and $s_{MSP}$ be the uncertainty score of KNN and MSP, respectively.  The double-score method uses the same h but its uncertainty score is $s(x) = s_{MSP}(x) + \mu\cdot  s_{KNN}(x)$ where $\mu$ is the parameter to be tuned. Please see description of double-score methods in Sec 2.4.
>
> R: “15. In Table 2, why is the selective risk at given TPR and FPR the same ...”
>
> A: We chose TPR=0.8 and used the ROC curves to select the smallest FPR attainable by all compared methods, i.e. FPR is determined by the method with the worst ID/OOD discriminator. Therefore the methods can easily satisfy this (conservative) requirement on FPR , and the selective risk is mainly influenced by the constraint on TPR=0.8 which is fixed for all OOD data. We did not show results for more settings of TPR/FPR due space limit.

---

> > ### Comment · Reviewer_hjaD · 2023-08-16
> > **Rebuttal acknowledgement**
> >
> > Thank you for the clarifications. I have read through all the reviews and rebuttals, and most of my questions/concerns have been resolved. It would help to address these points clearly in the revised paper (e.g., the fact that the practical use of the proposed work does not require exact OOD knowledge).

---

> > > ### Author Response · Authors · 2023-08-17
> > >
> > > Thank you for your time spent on our paper and for the thorough feedback we received. We will use your comments to improve the revised version of the paper.

---

### Author Rebuttal · Authors · 2023-08-09

We thank all the reviewers for their insightful feedback, which will significantly contribute to improving the paper's quality. Two primary objections were consistently raised across multiple reviews. In this global response, we aim to diligently address these objections.

Objection 1: The reviewers express concern regarding the assumption of possessing knowledge about the out-of-distribution (OOD) distribution, as it renders the proposed method impractical.

We understand your concern about the assumption. We would like to clarify that our aim is **not** to propose a novel algorithm dependent on OOD distribution knowledge for practical use. Rather, we use OOD distribution knowledge for precise problem formulation and deriving optimal strategies. The exact OOD distribution is unnecessary for practical use of our results:

Firstly, the problem formulation empowers the design of dependable evaluation metrics, essential for assessing and comparing out-of-distribution detection (OODD) techniques. It's noteworthy that during evaluation, while OOD remains unknown we possess a sample from both OOD and ID, such as CIFAR and MNIST. We empirically show that our proposed evaluation metrics yield a distinct method comparison, addressing the issues of existing evaluation metrics that concentrate on one aspect of the OOD classifier only and yield inconsistent rankings.

Secondly, knowledge of the optimal solution's structure enables us to narrow down the hypothesis class from which classifiers are learned. Specifically, our search can be confined to the class of double-score strategies, guided by the known desirable properties of these scores. Our experiments demonstrates that this approach, without introducing any new score, outperforms several existing methods that do not capitalize on this knowledge.

In summary, our utilization of OOD distribution knowledge is geared towards refining the problem formulation, enhancing the evaluation metrics, and narrowing the search space for classifier learning, rather than relying on explicit OOD distribution information for algorithmic execution.

Objection 2: The experiments are limited to relatively small datasets.

While we acknowledge the potential value of conducting extensive empirical evaluations on larger datasets, we would like to clarify that our paper's primary focus does not lie in this direction. Instead, our paper introduces a robust new methodology designed to facilitate such comparisons. The goal of the experiments is to prove that the theoretical concepts proposed in the paper are valid and can be readily applied in practice. Specifically, our intent is twofold:

i) To showcase that the newly proposed evaluation metrics offer a distinct perspective on the compared methods, and address limitations of existing metrics.

ii) To illustrate that even the presented straightforward application of the double-score concept yields superior performance based on the proposed metrics.

---

### Decision · Program_Chairs · 2023-09-21

**Decision:**

Reject

**Comment:**

The paper provides a unified framework for characterizing the optimal decision rule in settings with OOD data. In doing so, it extends the standard learning to reject setting, typically studied with ID samples, to handle both misclassified ID and OOD samples. Based on their theoretical findings, they propose a double-score OOD method and provide supporting experimental results.

The reviewers generally agree that problem addressed is timely, the optimal decision rules have an intuitive form, and the unified viewpoint presented is interesting. Two key concerns raised where about (i) the proposed method requiring access to the OOD distribution and (ii) the experimental setup being unclear (Reviewer msSt) and the experimental results being insufficient (Reviewer G4Mq). While the authors clarify that the exact OOD distribution is unnecessary for practical use of their results, they did not sufficiently address the second concern.

The following quote from Reviewer msSt summarizes the concern abouts the experimental results:
>  What models/backbones did the authors use in this framework? How did they tune the hyper-parameters for the methods? How were the models trained? How statistically significant are the results? How were datasets with inherently different shapes transformed to be processed by the same model? These questions are not answered  .... I understand that the experimental suite is not the main contribution of the paper but in its current form the experimental section adds very little value and does not substantiate the superiority claim of the proposed double-score strategy. As the paper proposes new evaluation metrics, I consider it important that these metrics are showcased on a well documented experimental setup.

Given the weak experimental section, we are unable to accept this paper in its current form.